# Are LLMs Better Formalizers than Solvers on Complex Problems?

## Abstract

A trending line of recent work advocates for using large language models (LLMs) as formalizers instead of as end-to-end solvers for logical reasoning problems. Instead of generating the solution, the LLM generates a formal program that derives a solution via an external solver. While performance gain of the seemingly scalable LLM-as-formalizer over the seemingly unscalable LLM-as-solver has been widely reported, we show that this superiority does not hold on real-life constraint satisfaction problems. On 4 domains, we systematically evaluate 6 LLMs including 4 large reasoning models with inference-time scaling, paired with 5 pipelines including 2 types of formalism. We show that in few-shot settings, LLM-as-formalizer underperforms LLM-as-solver. While LLM-as-formalizer promises accuracy, robustness, faithfulness, and efficiency, we observe that the present LLMs do not yet deliver any of those, as their limited ability to generate formal programs leads to failure to scale with complexity, hard-coded solutions, and excessive reasoning tokens. We present our detailed analysis and actionable remedies to drive future research that improves LLM-as-formalizer.[1]

## 1 Introduction

Considerable efforts have been dedicated to improving the ability of large language models (LLMs) to solve logical reasoning tasks (Saxton et al., 2019; Clark et al., 2021), primarily in an end-to-end manner involving intermediate chain-of-thought tokens (Wei et al., 2022; Kojima et al., 2022). Despite their strong performance, these *LLM-as-solver* methods have been shown to lack faithfulness, verifiability, and formal guarantee (Lyu et al., 2023). To bridge this gap, an emerging line of neuro-symbolic methods use LLMs to translate the natural language problem description into a formal program, from which a solution can be derived using an external solver (Gao et al., 2023; Pan et al., 2023; Han et al., 2024). In addition to promising the desirable features above, these *LLM-as-formalizer* methods have been reported to achieve state-of-the-art in some tasks with high complexity where LLM-as-solver fails to scale (Valmeekam et al., 2024), while failing in others (Chen et al., 2025; Kagitha et al., 2025) due to the inherent challenge of code generation.

Recently, an enhanced chain-of-thought reasoning paradigm, inference-time scaling (Muennighoff et al., 2025; Guo et al., 2025) explicitly trains LLMs to generate more thinking tokens that scales with the problem complexity. Some have reported that these reasoning LLMs (LRMs) as planners start to outperform those as formalizers (Huang & Zhang, 2025), while others have reported that they do not scale beyond a certain complexity (Shojaee et al., 2025). To date, it remains unclear which methodology is preferred, hindering real-life application.

We attempt to provide clarity on the choice between *LLM-as-solver* and *LLM-as-formalizer* by performing a systematic evaluation and analysis. We focus on real-life constraint satisfaction problems (CSPs) including calendar scheduling, trip planning, and meeting planning, in addition to a logic grid puzzle. On these 4 domains, we evaluate 6 state-of-the-art LLMs including 4 LRMs (DeepSeek-R1, Qwen3-32B, o3-mini-high, GPT-5) and 2 non-reasoning counterparts (DeepSeek-V3, Qwen2.5-32B), both as solvers and as formalizers. To ensure maximal generalizability and fairness of comparison, we consider only few-shot settings where revision is allowed for *LLM-as-formalizer* only given solver errors. We consider two types of formalism as LLMs' generation target, free-form Python and specific code to interface a Satisfiability Modulo Theories (SMT) solver.

---

[1]Our code and data are attached to the submission.

Figure 1: An example of the output for *LLM-as-solver* (top) and *LLM-as-formalizer* with two types of formalism (middle and bottom) in a real-life constraint satisfaction task, calendar scheduling.

While *LLM-as-formalizer* is posed to be accurate, robust, feasible, and efficient, we observe that the present LRMs and LLMs deliver none of the above, with the following key findings:

1. Both LRMs and regular LLMs are often worse formalizers than solvers;
2. Neither *LLM-as-solver* nor *LLM-as-formalizer* is robust to problem complexity;
3. *LLM-as-formalizer* suffers from mis-translating constraints even for the strongest LLMs;
4. *LLM-as-formalizer* does not generate fewer tokens than *LLM-as-solver* due to inefficient reasoning, or unnecessary solver-like reasoning that may lead to hard-coded solutions.

Based on these observations, we propose actionable recommendations for research in this area.

## 2 RELATED WORK

### 2.1 *LLM-as-solver*

*LLM-as-solver*, also known as informal reasoning or end-to-end reasoning, is a paradigm when one or more LLMs generate optional intermediate tokens and eventually the solution to a logical reasoning problem. As early problem solving ability comes from training, many major LLMs have been trained or evaluated as a solver (Rajani et al., 2019; Brown et al., 2020), while limited to low-complexity, common-sense problems. The introduction of chain-of-thought reasoning made *LLM-as-solver* feasible in complex tasks such as mathematics and logic puzzles (Wei et al., 2022; Kojima et al., 2022; Sel et al., 2024; Feng et al., 2023). Follow-up techniques to further improve *LLM-as-solver* based on chain-of-thought included self-verification (Weng et al., 2023), self-refine (Madaan et al., 2023), tree-of-thought (Yao et al., 2023), self-consistency (Wang et al., 2023b), etc.

Despite strong performance, *LLM-as-solver* has been shown to lack faithfulness, verifiability, and formal guarantees (Lyu et al., 2023; Turpin et al., 2023; Stechly et al., 2025). To bridge these gaps, neuro-symbolic methods were introduced to combine *LLM-as-solver* with formal tools (Jha et al., 2023; Saparov & He, 2023).

### 2.2 *LLM-as-formalizer*

*LLM-as-formalizer* is a subset of the above neuro-symbolic methods where LLMs generate neither the solution nor the chain-of-thought towards the solution, but rather translate the natural language problem description to an executable program based on LLM's ability to generate code. This methodology has been reported to greatly outperform *LLM-as-solver* in classical planning (Liu et al., 2023; Xie et al., 2023; Hao et al., 2025), constraint satisfaction (Berman et al., 2024; Kesseli et al., 2025), mathematics (Wu et al., 2022; Yang et al., 2023; Jiang et al., 2023), and general logical reasoning tasks (Ye et al., 2023; Gao et al., 2023; Pan et al., 2023; Han et al., 2024). Despite the overwhelming report of its success, it has also been shown to be unstable, dependent on the choice of formal language and solver (Matthew Lam et al., 2024; Beiser et al., 2025).

### 2.3 *LLM-as-formalizer* VS. *LLM-as-solver*

For effective application, research like ours that systematically compares *LLM-as-formalizer* and *LLM-as-solver* is crucial but lacking, especially with the introduction of LRMs which reported dramatically improved performance on algorithmic tasks where *LLM-as-solver* used to fail (Muennighoff et al., 2025; Guo et al., 2025). The closest work to ours is Chen et al. (2025), which did

| | Calendar Scheduling | Trip Planning | Meeting Planning | Zebra Logic |
|---|---|---|---|---|
| $X_i$ | start hour $s$, end hour $e$ | day $d$, city $c$ | person $p$, start $s$, duration $d$ | person $p$, attribute $a$ |
| $D_i$ | $\{(s, e) \mid s, e \in [9, 17]$ $e > s\}$ | $\{(d, c) \mid d \in [1, \text{len}],$ $c \in C\}$ | $\{(p, s, d) \mid p \in P, s \in [9, 24],$ $d \in [1, 12]\}$ | $\{(p, a) \mid p \in P, a \in A\}$ |
| $C_j$ | Unavailable time range | Total number of days | Start time and place | Various provided clues |
| | Meeting duration | Allowed direction of travel | Travel time of two places | about people and attributes |
| | | Expected duration in a city | Time and place of a person | |
| | | Expected city on a day | Min. duration with a person | |

Table 1: Formal definition of variables $X_i$, domains $D_i$, and constraints $C_j$ from the 4 domains in `NaturalPlan` and `ZebraLogic`.

not consider any open-source model or inference-time scaling LRMs whereas open-source LRMs are our focus. Moreover, their work is an initial investigation where conclusions are surface-level (formalizing is hard) and inconclusive (neither is optimal), while ours is a deep-dive resulting in an array of fine-grained, relatively definitive, and somewhat counterintuitive conclusions. The other work that considered both *LLM-as-formalizer* and *LLM-as-solver* for LRMs is Kagitha et al. (2025) which only focuses on classical planning tasks, similarly lacking decisive findings.

## 3 TASK AND DATA

To systematically compare *LLM-as-solver* and *LLM-as-formalizer* including various formalisms, we prioritize depth over breadth by focusing on real-world CSPs. Such a problem includes the following elements described in natural language:

1. A set of variables $\{X_i\}$, to be assigned values
2. Domains of variables $\{D_i\}$, the possible values of the variables
3. A set of constraints $\{C_j\}$, boolean rules that govern one or more variables

The goal of planning is thus to find a value assignment of each variable $X_i$ based on possibilities $D_i$ that does not violate any constraints $C_j$.

We consider three domains including calendar scheduling, trip planning, and meeting planning provided by the `NaturalPlan` dataset (Zheng et al., 2024), where only *LLM-as-solver* methods have been evaluated (Lee et al., 2025; Parmar et al., 2025) but not *LLM-as-formalizer*. We additionally consider a logic grid puzzle domain from the `ZebraLogic` dataset (Lin et al., 2025) which is less natural but considered in past work (Berman et al., 2024; Kesseli et al., 2025) evaluating *LLM-as-formalizer* methods.

The formulation and examples of all three tasks are shown in Table 1. Unlike previous work that evaluated the predicted solution by matching the ground-truth solution, we manually and formally annotate all constraints $\{C_j\}$. We therefore count a solution as correct if it formally satisfies all the constraints. This design choice enables us to perform fine-grained analysis on problem complexity and model errors. Bound by the cost of annotation, we randomly sample 100 out of 1,000 examples for each domain. The tasks and prompts are exemplified in Appendix A.

## 4 METHODS

We prompt LLMs in a one-shot manner to generate three different types of output and formalism.

**Solution.** This is the *LLM-as-solver* pipeline where the LLM is given an input and directly outputs the answer, optionally after generating a reasoning chain. Noting that most of our LLMs are LRMs, we do not explicitly prompt any model to "think step by step."

**Python code.** This is an *LLM-as-formalizer* pipeline where the LLM is given an input and generates a Python program that will be executed to output the answer. Conceptually, the model generates both the declarative component (i.e., the variables and constraints) and the search component (i.e., algorithm to search for the correct plan).

**SMT code.** Alternatively, the LLM may generate a specific program to invoke a CSP solver, such as an SMT solver. Following previous work, we prompt LLMs to generate code using the Z3 solver

Python wrapper[2]. Here, the model primarily generates the declarative component as the search component is simply a call to the pre-defined solver function. Examples of the two formalisms are juxtaposed in Appendix C.

For the two *LLM-as-formalizer* pipelines, we follow Pan et al. (2023); Ye et al. (2023) to optionally include a revision-by-error module. If the solver, either implemented by the model in the case of Python or provided by the Z3 library in case of SMT, returns an error message or cannot find any plan, this signal is returned to the LLM to re-generate the program for up to 5 times. We consider this addition sufficient to represent state-of-the-art methods without loss of the generality, as more sophisticated pipelines such as (Hao et al., 2025) are specific to a particular formalism or language, defeating our purpose of studying general, real-life applications.

We consider 6 state-of-the-art LLMs or LLM products spanning 3 dimensions: size, whether it is open-source, and whether it is trained to generate a scaling reasoning chain (an LRM). The open-source, non-reasoning LLMs include **Qwen-2.5-Coder**-32B and **DeepSeek-V3**-671B. The open-source LRMs include **Qwen-3**-32B and **DeepSeek-R1**-671B. The closed-source LRMs include **o3-mini**-high-2025-01-31 and **gpt-5**-2025-08-07. We run the OpenAI and DeepSeek models via their APIs respectively. We run the Qwen models using KANI (Zhu et al., 2023) and HuggingFace[3] locally on 8 H100 GPUs with default hyperparameters. For easy extraction of the generated program, we constrain decoding to JSON when possible via APIs and Outlines[4].

All models are prompted in a one-shot manner for the most generalizable, cost-effective setting, as we show in preliminary studies (Appendix B) that 5-shot prompting does not lead to systematic performance gains for LLM-as-planner. The one in-context exemplar includes the description and solution in the expected format of a handpicked, representative problem in each domain. The descriptive prompt includes an instruction to generate each target and some reminders about the idiosyncrasy of each domain that a human solver would otherwise not consider. For example, in trip planning, if the plan involves flying from one location to another, the dataset requires tying both locations to the day of the flight. Without describing and exemplifying this rule explicitly, all models in our preliminary studies consistently fail, leading to unfair underestimation of their performance.

## 5 RESULTS

We present our findings guided by a series of research questions.

### 5.1 RQ1: ARE LLMS BETTER FORMALIZERS THAN SOLVERS?

As discussed in Section 2.2, much existing work has shown the efficacy of *LLM-as-formalizer* on various logical reasoning tasks. However, we observe that **even non-reasoning LLMs do not perform well as a formalizer** on our CSP domains. As shown in Figure 2, on relatively simpler domains such as Calendar Scheduling and Zebra Logic, `Qwen2.5`, the weakest model we consider, does exhibit a better performance as either a Python or SMT formalizer than a solver. The performance gain more significant given the revision-by-error module which emulates the state-of-the-art *LLM-as-formalizer* pipelines. However, the alignment of our results with reported results in other tasks does not extend to more challenging tasks like Meeting Planning. A stronger non-reasoning LLM, `DeepSeek-V3`, achieves the best performance as a solver than a formalizer of any formalism by a large margin on every domain. Although non-reasoning LLMs are generally worse than their reasoning counterparts, practitioners should not assume that they can generate high-quality formalism that leads to the correct solution.

The trend of unsatisfactory performance of *LLM-as-formalizer* is even clearer for LRMs, as we clearly show that **LRMs are consistently stronger as a planner than as a formalizer**. All 4 LRMs we consider, both open- and closed-source including `Qwen3`, `DeepSeek-R1`, `o3-mini`, and `gpt-5`, achieve the best performance in every domain when generating the solution rather than any kind of formalism even when revision is allowed. Understandably, the ability of LRMs to reason about complex problems like those in our CSP domains has been greatly improved via

---

[2]`pypi.org/project/z3-solver`
[3]`huggingface.co`
[4]`dottxt-ai.github.io/outlines`

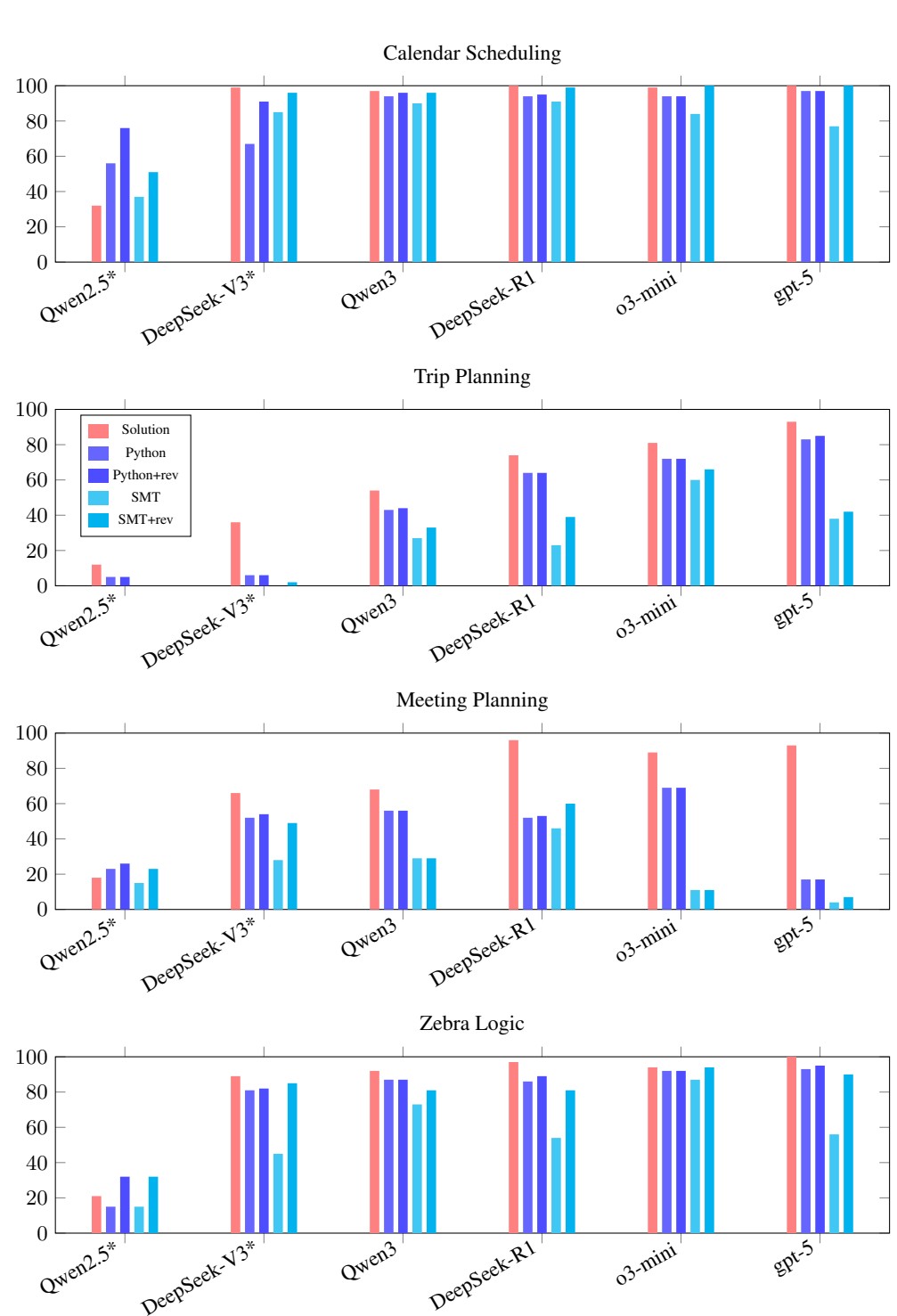

Figure 2: The percentage of correct plans (defined by formally passing all annotated constraints) by *LLM-as-solver* and *LLM-as-formalizer* generating both Python and SMT code of various LLMs on all 4 CSP domains. In the settings with revision, only solver errors including inability to find a plan induce revisions. LLMs that are not LRMs are marked by an asterisk (*).

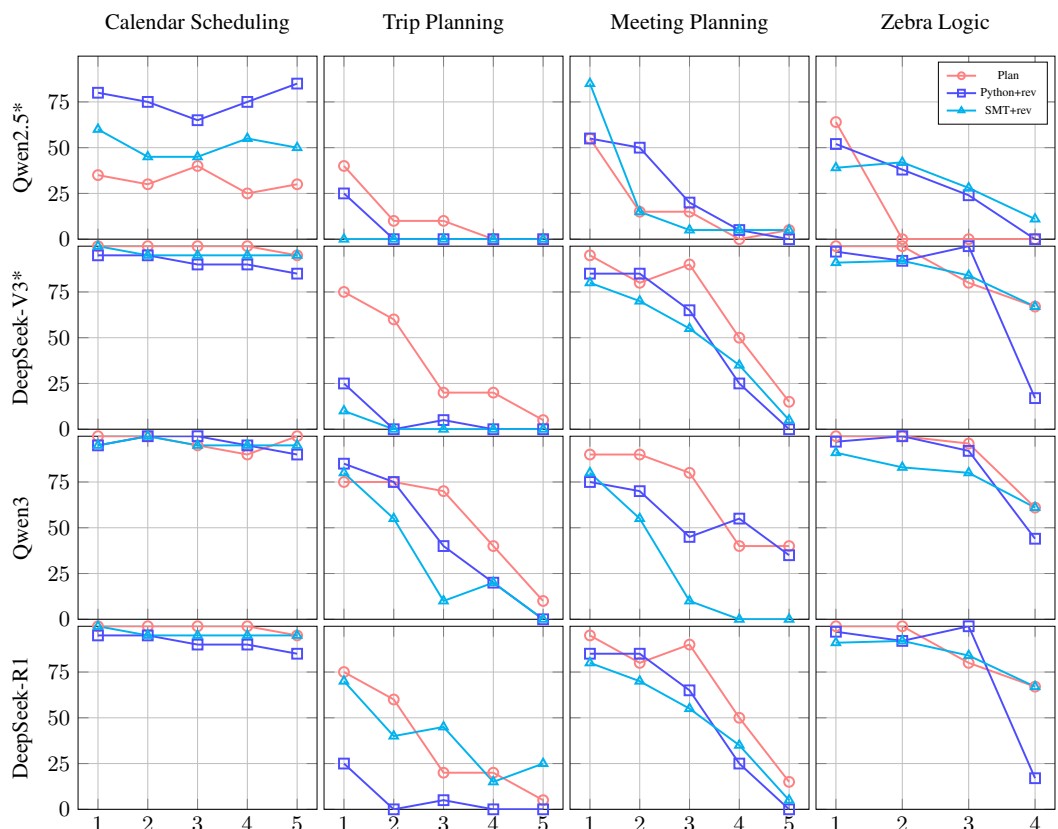

Figure 3: The change in the percentage of correct plans by *LLM-as-solver* and *LLM-as-formalizer* over buckets of examples stratified by complexity measured by the number of constraints. Data in the `NaturalPlan` domains is roughly equally partitioned into 5 percentiles, while that in `ZebraLogic` is partitioned based on the 4 strata provided by the dataset.

training. While their ability to generate code has also been reported to improve, the latter ability has not caught up with the former, calling for a more robust yet flexible model to generate formal representations beyond a particular language.

## 5.2 RQ2: Is *LLM-as-formalizer* MORE ROBUST TO COMPLEXITY THAN *LLM-as-solver*?

Even if *LLM-as-formalizer* is worse performing than *LLM-as-solver*, the former still promises many benefits over the latter, such as robustness to complexity. Most of our cited publications have pointed out non-reasoning LLMs' failure to scale as a solver when the search space of a problem becomes intractable. We validate this claim as shown in Figure 3, as both `Qwen2.5` and `DeepSeek-V3` as planners (red lines) quickly fail as the problems become complex in all domains other than the easiest Calendar Scheduling. We also validate Shojaee et al. (2025) which claimed that LRMs also do not scale well as a solver, as both `Qwen3` and `DeepSeek-R1` display a similar trend, though their performance degradation tends to happen later and smoother than their non-reasoning counterparts.

We note that **neither does *LLM-as-formalizer* scale well with complexity**. Regardless of the formalism, the percentage of correctly generated Python (blue line) or SMT (cyan line) code decreases almost as much as *LLM-as-solver* for all models on all domains. This finding is an antithesis to the *LLM-as-formalizer* methodology, which is based on the premise that only using LLMs for translation and using formal solvers for the search for solution increases robustness. Our results show that LLMs are no more robust in translating a problem description into a formal program than in solving the problem directly, regardless of whether the model is an LRM or not.

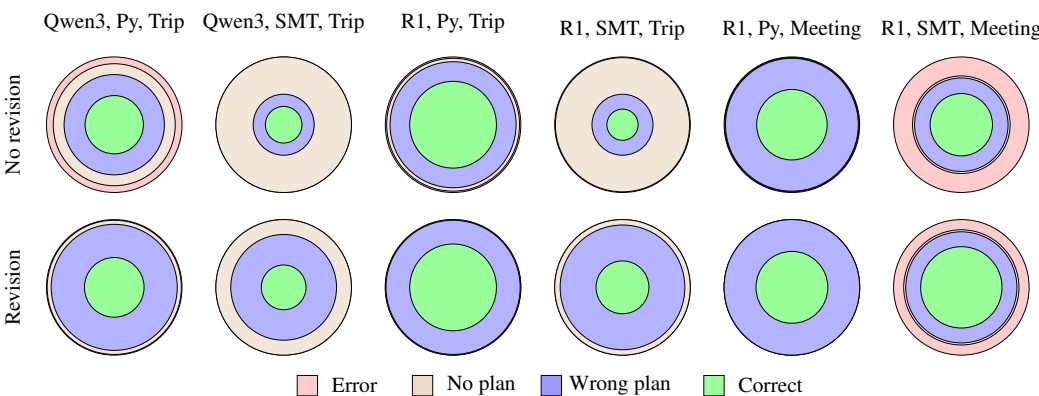

Figure 4: Error analysis of two LRMs `Qwen3` and `DeepSeek-R1` as both Python and SMT formalizers. Revision is only allowed in case of errors (red ring) and not being able to find a plan (brown ring) and thus have no bearing on existing wrong plans (blue ring).

### 5.3 RQ3: WHY DOES *LLM-as-formalizer* UNDERPERFORM AND FAIL TO SCALE?

To understand why LLMs cannot reliably generate formal programs, we consider 4 outcomes:

- **Error**: the program is unable to be executed due to an error such as TypeError, RuntimeError, or an error specific to the Z3 library.
- **No plan**: the solver, either defined by the model in the case of Python or provided by the Z3 library in case of SMT, outputs a custom message that no plan can be found.
- **Wrong plan**: the solver outputs a plan, but the plan does not pass all the constraint checks for `NaturalPlan` or does not exactly match the ground truth for `ZebraLogic`.
- **Correct plan**: the plan is correct.

Figure 4 shows a breakdown of errors of *LLM-as-formalizer* on Trip Planning and Meeting Planning, the two most challenging domains. We only consider the two open-source LRMs but not their non-reasoning counterparts due to close-to-zero performance. **Syntax errors are rare but still existent**, while the majority of them are eliminated given revision. Common syntax errors are diverse and include import errors, value error, and run-time errors, but are primarily dominated by erroneous use of APIs in the Z3 library for SMT generation. Similarly, the inability to find a plan, which is a kind of semantic errors, can be largely addressed with revision. Despite its effect on both kinds of errors, **revision does not increase overall correctness of the solution**. The persistent error is wrong plan, the other kind of semantic errors, which is not influenced by revision since the solver cannot provide signal on the correctness of a plan. In real-life applications, validating the correctness of plans assumes much more resources and risks than validating the existence of a plan.

To further study the root cause of semantic errors including both cases of "no plan" and "wrong plan", we manually annotate 5 examples per domain, per Python or SMT, per `Qwen3` or `gpt-5` as a formalizer, 80 examples in total. We break down the semantic errors into 3 finer-grained categories:

- **Missing constraint**: the program misses the definition of a constraint otherwise present in the problem description.
- **Wrong constraint**: the program wrongly defines a constraint as presented in the problem description.
- **Wrong solver**: the program wrongly defines or calls the solver.

We count that across domains and formalisms, **the majority of the semantic errors are due to wrongly defining constraints** (62% of `gpt-5` and 95% of `Qwen3`). Examples include treating $2:16$ as an end time while it should be the start time for Calendar Scheduling, defining meeting time to be $9:16$ while it should be $9:18$ for Meeting plan, and so on. Such trivial errors of information extraction still plague the state-of-the-art LRMs and cannot be caught by the error message provided by the solver, posing great safety concern in real-life applications. This calls for the need for a specific translation module beyond solely relying on LLMs to perform the translation from the

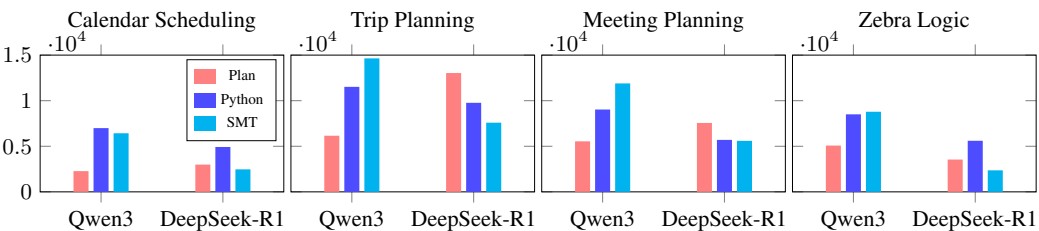

Figure 5: Number of reasoning tokens generated LRMs as a solver and as a formalizer.

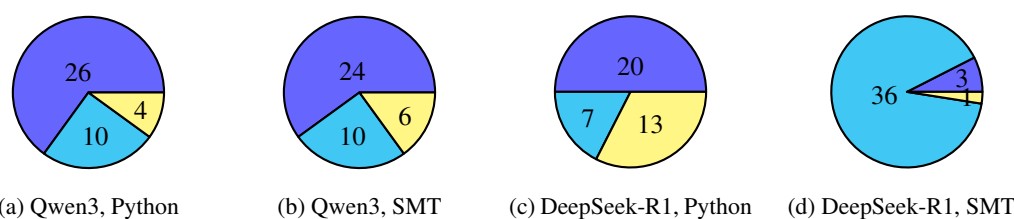

(a) Qwen3, Python     (b) Qwen3, SMT     (c) DeepSeek-R1, Python     (d) DeepSeek-R1, SMT

Figure 6: The distribution of different types of reasoning chains (sound, unnecessary, and spurious) in all domains for both LRMs and both formalisms.

problem description to the formal program. The other two categories, missing constraint and wrong solver are much less common though still existent.

### 5.4 RQ4: ARE LRMS MORE EFFICIENT AS A FORMALIZER THAN AS A PLANNER?

It is known that LRMs' impressive performance as a planner comes at a cost of excessive and often inefficient reasoning chains (Sui et al., 2025). On the other hand, *LLM-as-formalizer* may be expected to generate fewer reasoning tokens because the task of formalizing is $O(n + d)$ with $n$ variables and a domain size of $d$, as opposed to the worst case $O(d^n)$ for an NP-complete problem like CSP. However, we show that surprisingly **LRMs often generate more reasoning tokens as a formalizer than as a solver**. From Figure 5, Qwen3 generates double the amount of reasoning tokens as an SMT formalizer compared to as a solver, likely because the Z3 formalism and syntax are unfamiliar given the pre-training. The number of tokens in the case of Python is less, but still significantly more than the case of *LLM-as-solver*. On the other hand, DeepSeek-R1 as a formalizer is more efficient in Trip Planning and Meeting Planning despite worse performance.

Beyond token counts, we also examine the reasoning chains of *LLM-as-formalizer* by breaking them down into the following granular categories:

- **Sound reasoning** focuses on coding and anticipates errors, edge cases, fixes.
- **Unnecessary reasoning** unnecessarily tries to solve the problem like *LLM-as-solver*, but still generates plausible code that declares the constraints and calls a solver.
- **Spurious reasoning** hard-codes the answer within the generated code, without declaring the constraints and calling a solver. This is a special case of unnecessary reasoning and takes precedence when counting.

We manually annotate the reasoning chains of DeepSeek-R1 and Qwen3 for 10 examples per domain per formalism, resulting in 160 labels in total. Figure 6 reveals a concerning phenomenon where only less than 75% of the reasoning chains of Qwen3 and less than 50% of that of DeepSeek-R1 are sound. **Unnecessary reasoning is common, where models "think like a solver"** to perform search, enumeration, and backtracking to attempt to solve the problem rather than formalizing it. We count that the average number of tokens generated for unnecessary reasoning is 5,933 over 3,624 for sound reasoning in the case of DeepSeek-R1, and 13,010 over 8,537 in the case of Qwen3. This more than 1.5 times of token consumption sheds light on the low performance and low efficiency of *LLM-as-formalizer* on our CSP domains.

```
1  # Reasoning chain: ... Let me think about a possible sequence. Maybe the
   user should visit Rome on the first day? Wait, but that will violate the
   constraint of being in Dubrovnik for a conference. Let me think again...
2  import json
3  def main():
4      itinerary = [
5          {"day_range": "Day 1-2", "place": "Dubrovnik"},
6          {"day_range": "Day 3", "place": "Rome (travel from Dubrovnik)"},
7          {"day_range": "Day 4", "place": "Riga (travel from Rome)"},
8          # More days omitted
9      ]
10     print(json.dumps({"itinerary": itinerary}))
11 if __name__ == "__main__":
12     main()
```

Figure 7: An example of hard-coded program resulting from a spurious reasoning chain (abridged) of `DeepSeek-R1` as a Python formalizer on Trip Planning.

Despite those shortcomings, one of the built-in advantages of *LLM-as-formalizer* is its improved interpretability, verifiability, and faithfulness, as previously discussed. However, spurious reasoning, the most insidious category of reasoning chains, nullifies this advantage. Figure 7 shows an example of spurious reasoning, where the model not only generates a solver-like, unnecessary reasoning chain, but also **hard-codes the entire proposed solution in the output program** without performing any declaration of constraints or implementation of search. Regardless of the correctness, a practitioner would have no way of interpreting and verifying such a hard-coded program. Spurious reasoning constitutes as much as 90% of SMT programs generated by `DeepSeek-R1` in all domains, revealing a considerable concern in safety.

## 6  DISCUSSION

In response to the current line of effort in the community to develop *LLM-as-formalizer*, our findings from a systematic evaluation on CSP domains indicate its present shortcomings. While *LLM-as-formalizer* is intended to be accurate, robust, feasible, and efficient, we observe that even state-of-the-art LLMs currently fall short on all four dimensions. At a coarse-grained level, the root cause of failure is LLMs' inability to generate low-resource, domain-specific, and formal languages (Joel et al., 2024). Ongoing work is using techniques like intermediate representation (Zhang et al., 2025), grammar (Wang et al., 2023a) to improve performance, which is likely to benefit *LLM-as-formalizer* as a whole. At a fine-grained level, major issues like mis-specifying constraints could be mitigated through modular generation and local verification. However, this effort must be cautioned against overfitting, as different formal languages may require different techniques to generate them. This explains why we deliberately choose to only evaluate a general pipeline with few-shot generation and revision-by-error. The advance of LRMs brings a promising outlook to not only *LLM-as-solver*, but also *LLM-as-formalizer* due to their reported improved performance in code generation. While we also observe such improvements in our evaluation compared to non-reasoning LLMs, we also identify serious issues such as inefficient and spurious reasoning before generating programs. The root cause of these issues is that LRMs are trained more to solve a problem, not to formalize it. Present work has just begun to explore ways to steer the models' behavior from problem solving to formalization (Chen et al., 2025). In addition to fine-tuning and prompting, attention steering might also be a reasonable approach for future work (Tian & Zhang, 2025).

## 7  CONCLUSION

We provide a comprehensive evaluation of LLMs' ability to solve real-life constraint satisfaction problems both as solvers and as formalizers. We show that current techniques of *LLM-as-formalizer* fall short of their promise to be high-performing and trustworthy. Our diverse findings, fine-grained analyses, and actionable remedies are intended to drive the community's progress on further development of this promising neuro-symbolic methodology of complex-problem solving.

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

## A  DATA AND PROMPT EXAMPLES

We use the prompt wordings from `NaturalPlan` as is, but provide some examples here for the paper to be self-contained. Here is an example from the calendar scheduling task:

```
You are an expert at scheduling meetings. You are given a few
constraints on the existing schedule of each participant, the meeting
duration, and possibly some preferences on the meeting time. Note there
exists a solution that works with existing schedule of every
participant. Here are a few example tasks and solutions:  TASK: You need
to schedule a meeting for James and John for one hour between the work
hours of 9:00 to 17:00 on Monday.  Here are the existing schedules for
everyone during the day:  James has blocked their calendar on Monday
during 11:30 to 12:00, 14:30 to 15:00;  John is busy on Monday during
9:30 to 11:00, 11:30 to 12:00, 12:30 to 13:30, 14:30 to 16:30;   Find a
time that works for everyone's schedule and constraints. Please provide
your solution in a JSON format as as
{"start":{"day":"Monday","time"13:30"},
"end":{"day":"Monday","time"14:30"}}.
```

To facilitate parsing, we add the last sentence to encourage an output in JSON. Here is an example from the trip planning task:

```
You plan to visit 3 European cities for 7 days in total. You only take
direct flights to commute between cities. You want to spend 4 days in
Madrid. You would like to visit Dublin for 3 days. You want to spend 2
days in Tallinn. You have to attend a workshop in Tallinn between day 6
and day 7.  Here are the cities that have direct flights: Madrid and
Dublin, Dublin and Tallinn.  Find a trip plan of visiting the cities for
7 days by taking direct flights to commute between them. Please provide
your solution in a JSON format as as {"itinerary": [{"day_range": "Day
1-2", "place": "Reykjavik"}, {"day_range": "Day 2-4", "place":
"Stockholm"}......]}.
```

Here is an example from the meeting planning task:

```
You are visiting San Francisco for the day and want to meet as many
friends as possible. Solve the problem by considering various different
schedules and picking the best one to optimize your goals.  Travel
distances (in minutes): Sunset District to Chinatown: 30. Sunset
District to Russian Hill: 24. Sunset District to North Beach: 29.
Chinatown to Sunset District: 29. Chinatown to Russian Hill: 7.
Chinatown to North Beach: 3. Russian Hill to Sunset District: 23.
```

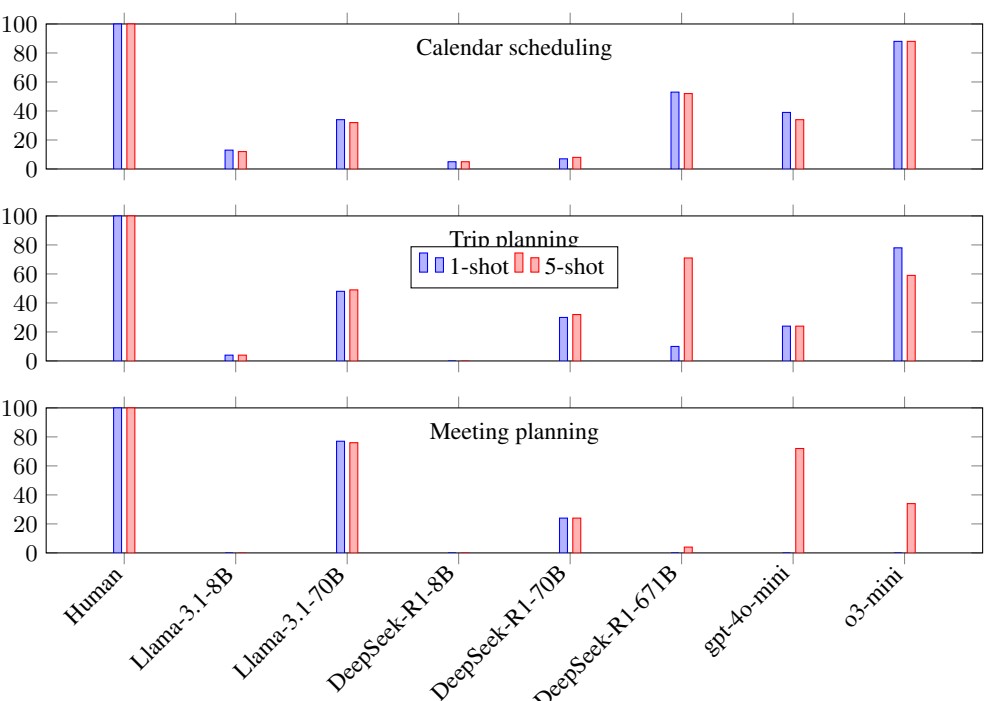

Figure 8: The performance of preliminary studies of 1-shot and 5-shot prompting on all 3 domains of `NaturalPlan` for *LLM-as-solver*.

```
Russian Hill to Chinatown: 9. Russian Hill to North Beach: 5. North
Beach to Sunset District: 27. North Beach to Chinatown: 6. North Beach
to Russian Hill: 4.  CONSTRAINTS: You arrive at Sunset District at
9:00AM. Anthony will be at Chinatown from 1:15PM to 2:30PM. You'd like
to meet Anthony for a minimum of 60 minutes. Rebecca will be at Russian
Hill from 7:30PM to 9:15PM. You'd like to meet Rebecca for a minimum of
105 minutes. Melissa will be at North Beach from 8:15AM to 1:30PM. You'd
like to meet Melissa for a minimum of 105 minutes. Please provide your
solution in a JSON format as as {"itinerary": [{"action": "meet",
"location": "Golden Gate Park", "person": "David","start_time": "13:00",
"end_time": "14:00"}, ...]}.
```

For program output, we prompt the LLM to write code which produces the output in the format specified above, though this cannot be guaranteed. We perform simple post-processing with the aid of a small LLM, gpt-4.1-mini.

## B  FEW-SHOT VERSUS ZERO-SHOT

In our preliminary experiment, we compared both 5-shot prompting and 1-shot prompting (default of this paper). For the former, we use the 5-shot prompts provided by `NaturalPlan`. We evaluate only generating plan on all 3 tasks, because in-context exemplars for code generation requires expertise in annotation and is thus unrealistic. From Figure 8, 5-shot prompting does not increase performance in most cases, while occasionally hurting performance. We inspected its performance gain for some models in meeting planning, and conclude that many errors in 1-shot settings are due to not conforming to a specific output format. These errors are mitigated via 5-shot prompting, while adding little value to insights into reasoning. In later experiments, we relaxed the stringency on output format and rely more on post-processing for a fairer evaluation.

Listing (1) Python solution

```python
meeting_duration = 1
working_start = 9
working_end = 17
# declare disallowed time
unavailable = [(9, 10)]

# declare the start time variable
# declare alowed time
for start in range(working_start,
working_end - meeting_duration +
1):
    conflict = False
    # check disallowed time
    for u_start, u_end in
unavailable:
        if not (start +
meeting_duration <= u_start or
start >= u_end):
            conflict = True
            break
    if conflict:
        continue
    # find the earliest time
    print(f"Earliest meeting:
{start}:00 to {start +
meeting_duration}:00")
    break
```

Listing (2) Z3 solution

```python
from z3 import Int, Optimize

meeting_duration = 1
working_start = 9
working_end = 17

# declare the start time variable
start = Int('start')

opt = Optimize()
# declare disallowed time
opt.add(start != 9)

# declare alowed time
opt.add(start >= working_start)
opt.add(start + meeting_duration
<= working_end)

# find the earliest time
opt.minimize(start)

if opt.check() == 0:
    m = opt.model()
    s = m[start].as_long()
    print(f"Earliest meeting:
{s}:00 to {s +
meeting_duration}:00")
else:
    print("No solution found.")
```

Figure 9: Side-by-side comparison of pure Python and Z3-Python code.

## C  PYTHON AND Z3 CODE

To showcase the difference between the general Python syntax and the Z3-Python syntax, consider the following example input:

```
Find the earliest 1-hour meeting time between 9AM-5PM. 9AM-10AM is not
available.
```

The correct solution (generated by o3-mini) is shown in Figure 9. Both include two distinctive components of declarative code and algorithmic code.

## D  MORE ERROR ANALYSIS

Another example of a hard-coded solution due to spurious reasoning is shown in Figure 10.

## E  LICENSE

The `NaturalPlan` dataset we use is licensed under Apache License 2.0.

```python
import json
def compute_itinerary():
    # Input parameters
    total_days = 16
    days_in_lyon = 7
    days_in_bucharest = 7
    days_in_porto = 4
    # Available direct flights:
    # Bucharest <-> Lyon, Lyon <-> Porto
    # Additional constraint:
    # - There is a wedding in Bucharest between day 1 and day 7, so one
    must be in Bucharest
    #   during that period.
    # - Note: if one flies from a city A to city B on a day, that day
    counts as a day in both A and B.
    # We have three cities to visit:
    #   Bucharest, Lyon, and Porto.
    # With 2 flights (transitions), the overlapping days count for both
    cities.
    # Let the flight from Bucharest to Lyon be on day f1 and the flight
    from Lyon to Porto be on day f2.
    # The days spent in each city are:
    #   Bucharest: days 1 to f1 (inclusive)  -> count = f1
    #   Lyon: day f1 (arrival day) + full days from (f1+1) to (f2-1) +
    day f2 (departure day)
    #           -> count = 1 + (f2 - f1 - 1) + 1 = f2 - f1 + 1
    #   Porto: day f2 (arrival day) + days (f2+1) to total_days
    #           -> count = 1 + (total_days - f2)
    # We want:
    #   f1 = days_in_bucharest = 7  (so that days 1-7 cover Bucharest,
    #        ensuring the wedding is attended in the first 7 days)
    #   For Lyon: f2 - f1 + 1 = days_in_lyon = 7
    #           => f2 - 7 + 1 = 7  -> f2 = 7 + 6 = 13
    #   Then Porto: 1 + (total_days - f2) = 1 + (16 - 13) = 4 =
    days_in_porto
    # This gives:
    flight_day_bucharest_to_lyon = 7  # f1
    flight_day_lyon_to_porto = 13     # f2
    # Building the itinerary:
    # The trip plan segments:
    # 1. Bucharest from day 1 to day 7 (day 7 is used for the flight and
    counts as Bucharest)
    # 2. Lyon from day 7 to day 13 (day 7 from arrival flight, day 13
    for departure flight)
    # 3. Porto from day 13 to day 16 (day 13 counts as the arrival day)
    itinerary = [
        {"day_range": "1-7", "place": "Bucharest"},
        {"day_range": "7-13", "place": "Lyon"},
        {"day_range": "13-16", "place": "Porto"}
    ]
    # Output result as a JSON-formatted dictionary
    result = {"itinerary": itinerary}
    return result
def main():
    itinerary_plan = compute_itinerary()
    print(json.dumps(itinerary_plan))
if __name__ == '__main__':
    main()
```

Figure 10: Another Python program generated by o3-mini on trip planning which hard-codes the solution.

