# OpenReview forum: "Are LLMs Better Formalizers than Solvers on Complex Problems?"
_ICLR.cc/2026/Conference — Submitted to ICLR 2026_

### Official Review · Reviewer_mhMi · 2025-10-18

**Soundness:** 3
**Presentation:** 2
**Contribution:** 3
**Rating:** 8
**Confidence:** 4

**Summary:**

This paper analyzes the model’s performance and behavior when serving as a solver or formalizer, aiming to better understand its real-world applicability. The results demonstrate that using the model as a solver yields the best performance, robustness, and efficiency. Further analysis identifies the failure modes of the LLM-as-formalizer paradigm, providing valuable insights for strengthening this approach in future research.

**Strengths:**

- The paper is highly interesting and presents insightful findings.

- The experiments are well-designed and intuitive.

- The presentation and writing are clear, structured, and easy to follow.

**Weaknesses:**

- The analysis could be expanded to provide deeper insights. For instance, solvers such as Isabelle offer detailed feedback on why an execution fails—this kind of fine-grained diagnostic information could be more informative than the primarily syntactic feedback from Z3.

- Additionally, in Figure 2, Python appears to outperform SMT in many cases. This is a particularly intriguing finding, given that SMT solvers are specifically designed for constraint satisfaction tasks, whereas Python is a general-purpose programming language. Could Python potentially replace domain-specific languages in this context, considering that modern LLMs/LRMs are trained on large volumes of code? Expanding on this analysis could significantly strengthen the paper.

- Figure 4 appears overly complex and difficult to interpret. It might be clearer if presented in the same format as Figure 6.

**Questions:**

Please see the weaknesses.

---

> ### Author Response · Authors · 2025-11-19
> **Expansion of analysis**
>
> We thank the reviewer for the favorable remark and constructive suggestions. Indeed, a more informative solver is worth considering (also see discussions with reviewer p6Ft) while possibly diminishing performance if the formalism is lower-resource. This dovetails into the second point that LLMs/LRMs are shown to be stronger at formalizing general, high-resource languages rather than specific, low-resource languages. In fact, the same outcome was contemporaneously observed and studied in the neighboring field of classical planning [1]. Regrettably, we are unable to include those expansions due to scope limit but will be happy to do so in future work.
>
> [1] Unifying Inference-Time Planning Language Generation

---

> ### Author Response · Authors · 2025-11-19
> **Unclear figure**
>
> We agree and will adjust as suggested.

---

> > ### Comment · Reviewer_mhMi · 2025-11-26
> >
> > Thanks for your response. What about the second weakness—the lack of analysis on why Python outperforms SMT? Do you have any insights on this?

---

> > > ### Author Response · Authors · 2025-11-26
> > > **Why Python outperforms SMT?**
> > >
> > > We agree with the reviewer that this is indeed a topic that warrants further discussion. First, past work has shown that LLMs  are generally better at generating high-resource programming languages than low-resource ones [1]. In our work, another layer of complexity is added to this claim since the low-resource language (SMT) is better suited to express the CSP tasks. To dive deeper into the performance difference between Python and SMT, we will now examine Meeting Planning, a domain where the difference is the largest. (Note: in simpler domains like Calendar Scheduling and ZebraLogic, SMT performs on par with Python)
> > >
> > > From Figure 4, it is clear that for a model like DeepSeek-R1, the primary reason why formalizing SMT underpeforms formalizing Python is syntax errors (the red ring). This is also corroborated by past work like [1] as LLMs are significantly worse at generating the correct syntax of low-resource languages in general.
> > >
> > > To further verify this observation, we performed manual inspection of two additional models's failure cases, Qwen3-32B and gpt-5, formalizing Python and SMT, on Meeting Planning with the following results. (Note: these results already exist but we decided not to include them in the paper, but will do so after this discussion with the reviewer)
> > > |              | syntax error | semantic error |
> > > |--------------|--------------|----------------|
> > > | gpt-5-python | 2            | 3              |
> > > | gpt-5-smt    | 5            | 0              |
> > > | qwen3-python | 1            | 4              |
> > > | qwen3-smt    | 4            | 1              |
> > >
> > > A syntax error in generated Python is often related to encoding that may be fixed by post-processing, for example:
> > > > Using single quotes for a dictionary key containing an apostrophe — "Fisherman's Wharf" — which prematurely closed the string and triggered a SyntaxError.
> > >
> > > On the other hand, a syntax error in generated SMT code is often more involved, for example:
> > > > The program crashed with Z3Exception: Symbolic expressions cannot be cast to concrete Boolean values, because it attempted to use a Z3 expression (fi) in a normal Python if statement. In Z3, model values are symbolic objects until explicitly converted, so testing if fi != -1: is illegal.
> > >
> > > Figure 2&4 show that even with revision, only some of these syntax errors can be fixed. This convincingly concludes that formalizing Python outperforms SMT mostly due to _rather complicated_ syntax errors in the latter.
> > >
> > > Finally, we add that semantic errors exist in both formalisms and are rather consistent in frequency (e.g., missing constraints, wrong constraints, wrong output logic...); see Section 5.3.
> > >
> > > [1] Knowledge Transfer from High-Resource to Low-Resource Programming Languages for Code LLMs

---

> > > > ### Comment · Reviewer_mhMi · 2025-11-27
> > > >
> > > > Thanks. I recommend incorporating this into your revision. I think it would likely be of interest to the community.

---

### Official Review · Reviewer_R8gZ · 2025-10-19

**Soundness:** 3
**Presentation:** 3
**Contribution:** 3
**Rating:** 4
**Confidence:** 3

**Summary:**

This paper provides a systematic evaluation of the "LLM-as-formalizaer" paradigm. This paper tries to answer an interesting question -- "is LLM-as-formalizer better or LLM-as-solver better?" With focusing on CSP domain, they found LLM-as-formalizer underperforms LLM-as-solver in many tasks; formalizers are not robust; and fomarl language easily prone to erros (parsing or missing constraints).

**Strengths:**

- Interesting research question
- Comprehensive evaluation -- 6 LLMs, 4 domains, and 2 formal language
- FIne-grained analysis

**Weaknesses:**

- My biggest concern is the domain breath -- Im not sure if CSP is the best problem for comparing LLM-as-solver and LLM-as-formalizer. Some papers, for example [1], have studied the relationship between solver and formalizer by using a routing-based method. I feel like CSP are possibly not suitable to using symbolic/mathematical logic. Have you tried evaluation on logical / mathematical datasets?
- LLM-as-formalizer might be too simplified, have you tried more advanced LLM-as-formalizer methods?

I'm happy to raise my score if you can answer my biggest concern well : )

[1] Han, S., Liu, T., Li, C., Xiong, X., & Cohan, A. (2024). HYBRIDMIND: Meta Selection of Natural Language and Symbolic Language for Enhanced LLM Reasoning. arXiv preprint arXiv:2409.19381.

**Questions:**

See weakness

---

> ### Author Response · Authors · 2025-11-19
> **Choice of datasets and domains**
>
> The short answer is that we identify **first** the problem to solve (real-life CSP) and **then** the appropriate tool (Python or Z3). Overall, we agree with the assessment that LLM-as-formalizer is more effective in prior datasets than in ours (i.e, Meeting Planning, Trip Planning and Calendar Planning). We have made the same claim in line 34-36 while citing works in the planning domain. However, the key research question of our work is not “**on which tasks** is LLM-as-[Z3/any solver]-formalizer the most effective”, but rather “**whether** LLM-as-formalizer is effective on real-life CSP tasks” (line 46). What we did not emphasize enough is the motivation for those tasks themselves [1,2,3]. The datasets we consider are thus much linguistically richer, requiring common-sense inference along with algorithmic calculation. While we do not directly compare with logical reasoning datasets of a more synthetic nature (like ProofWriter), Section 5.3 shed light on translation errors that may be more rampant when the problem description is more natural.

---

> ### Author Response · Authors · 2025-11-19
> **More advanced LLM-as-formalizer approaches**
>
> While it is fair to suggest a more sophisticated LLM-as-formalizer approach, we argue that current work has not agreed upon what such an approach is. For example, the paper the review cited and [4] that route between LLM-as-solver and LLM-as-formalizer is not purely an LLM-as-formalizer approach and defeats the purpose of analyzing and comparing the two. On the other hand, existing work that optimizes SoTA LLM-as-formalizer such as [1,5] primarily employs a) a modular pipeline including chain-of-thought and b) revision by error. In our work, we have considered both. Finally, we’d like to re-emphasize the empirical value of evaluating commonly used techniques in literature, even though they may not contain substantial algorithmic advances.

---

> ### Author Response · Authors · 2025-11-19
> **References**
>
> [1] Planning Anything with Rigor: General-Purpose Zero-Shot Planning with LLM-based Formalized Programming (ICLR 2025)
>
> [2] TravelPlanner: A Benchmark for Real-World Planning with Language Agents (ICML 2024)
>
> [3] NATURAL PLAN: Benchmarking LLMs on Natural Language Planning
>
> [4] Steering Large Language Models between Code Execution and Textual Reasoning (ICLR 2025)
>
> [5] Unifying Inference-Time Planning Language Generation

---

### Official Review · Reviewer_jBmf · 2025-10-29

**Soundness:** 2
**Presentation:** 3
**Contribution:** 2
**Rating:** 4
**Confidence:** 4

**Summary:**

The paper studies the effectiveness of  large language models (LLMs) on planning tasks when they are used as formalizers (translating problem descriptions into formal programs) vs when they are used as solvers (directly generating answers).
The paper presents a series of empirical studies to investigate the research question. Specifically, the paper evaluates six models (including DeepSeek-R1, Qwen-3, and GPT-5) across four constraint satisfaction domains (such as trip planning). It analyzes accuracy, robustness to complexity, faithfulness, and efficiency. The paper draws the conclusion that current LLMs often underperform as formalizer, and do not deliver "accuracy, robustness, faithfulness, and efficiency”

**Strengths:**

The paper studies an important and interesting question to evaluate the advantage of LLMs as formalizers versus solvers. This has been a hot topic in reasoning research.

The paper is well-written and easy to understand, with well-organized research questions.

The experimental study covers multiple domains, models, and covers using both Python and SMT.

**Weaknesses:**

I appreciate the authors’ efforts in clearly organizing the research questions and results. However, I have some concerns regarding the interpretation of the findings.

In particular, for RQ3 (Section 5.3), the paper argues that LLMs-as-formalizers are unreliable (as also stated in the abstract) and provides a detailed analysis of their failure modes. However, it does not include an analysis of failures for LLMs-as-solvers, which makes the comparison incomplete.

While Section 5.3 touches faithfulness, it misses discussion of another important aspect of using LLMs-as-formalizers: LLM-as-formalizers may better refuse to give you an answer when it fails, whereas LLMs-as-solvers are more likely to produce an incorrect final answer. In this sense, LLMs-as-formalizers may be better calibrated or better in selective prediction behavior (it is better to abstain than give a wrong answer). Figure 4 suggests that SMT-based methods frequently fail to produce any solution (“no plan”). It would therefore be helpful to report the answer extraction rate for both paradigms and the accuracy normalized by the subset of cases where a final answer is produced.

The paper could also benefit from more discussion of problem complexity and scalability. As shown in Figure 3, many examples involve relatively few constraints, and Figure 5 indicates that the reasoning token counts are mostly within ~10K. This range may be insufficient to draw comprehensive conclusions about scalability. A more detailed analysis of performance versus reasoning tokens would strengthen the paper’s claims.

**Questions:**

Could the authors provide the answer extraction rate for the LLM-as-solver setting?

---

> ### Author Response · Authors · 2025-11-19
> **Lack of analysis of LLM-as-solver**
>
> While we agree with the reviewer that analysis with both methodologies improves completeness, we deliberately focus on just LLM-as-formalizer given the scope limitation for two explicit reasons. First, we have cited a similar analysis [1] for LLM-as-solver which does not touch on many idiosyncrasies of LLM-as-formalizer such as syntax and semantics in Section 5.3. Second, in recent literature, the two methodologies have been habitually studied separately. For example, in the classical planning domain, [2] and [3] are contemporaneous surveys for each of the methodology, each including papers that primarily study one over another. That said, we argue that it is sufficient to treat LLM-as-solver as a baseline as we did. Without needing tantamount analysis, we can still convincingly draw the conclusions as we have. We are happy to discuss further if needed.

---

> ### Author Response · Authors · 2025-11-19
> **Accuracy within attempt vs. abstain**
>
> We thank the reviewer for this interesting and informative suggestion. In all our experiments, the answer extraction rate for LLM-as-solver is always 100%, whereas that for LLM-as-formalizer is as reported in Figure 4 ranging from 30% to 100% (the sum of “wrong plan” and “correct”). Considering only the subset where a solution is attempted:
> | %                    | sol. | Py | Z3 |
> |----------------------|------|----|----|
> | R1-Calendar Planning | 100  | 95 | 99 |
> | R1-Trip Planning     | 74   | 69 | 51 |
> | R1-Meeting Planning  | 96   | 53 | 67 |
> In the simpler calendar planning task, the answer extraction rate is also close to 100% leading to little difference in conclusions. In harder tasks like trip and meeting planning, Z3 which _abstains_ more frequently sees an improvement in accuracy. In this case, it still underperforms LLM-as-solver but outperforms LLM-as-Python-formalizer in some cases. We will gladly add this analysis to our paper though it does not change our conclusions.

---

> ### Author Response · Authors · 2025-11-19
> **Scale bigger**
>
> We agree with the general need to “go bigger” in the scaling experiments. Realistically, we are limited by what the datasets provide as we consider augmenting existing datasets out-of-scope, while those suggested by reviewer p6Ft are even smaller in scale. We do want to emphasize that our choice to evaluate on ZebraLogic [4] is exactly for this purpose. Though its paper shows its scaling is beyond models at the time of its publication, it may eventually be insufficient for SoTA scaling LRMs. We hope the reviewer would agree that it is fair to leave to future work constructing datasets of even more problem complexity.

---

> ### Author Response · Authors · 2025-11-19
> **References**
>
> [1] The Illusion of Thinking: Understanding the Strengths and Limitations of Reasoning Models via the Lens of Problem Complexity
>
> [2] PlanGenLLMs: A Modern Survey of LLM Planning Capabilities
>
> [3] LLMs as Planning Formalizers: A Survey for Leveraging Large Language Models to Construct Automated Planning Models
>
> [4] ZebraLogic: On the Scaling Limits of LLMs for Logical Reasoning

---

### Official Review · Reviewer_p6Ft · 2025-11-01

**Soundness:** 2
**Presentation:** 3
**Contribution:** 2
**Rating:** 4
**Confidence:** 4

**Summary:**

The authors rigourously explore whether LLMs as a formalizer where the LLM translates the NL input to language suitable to the solver is effective over LLM as a solver, where an LLM is directly used to solve the problem.

**Strengths:**

1. The paper is thorough with interesting experiments to show that LLMs/LRMs as a formalizer paradigm still suffers from major lacunae i.e, with increase in the problem complexity the performance of the LLMs becomes considerably low.
2. The paper is well presented with appendix consisting of all examples of prompts/generated code by the LLMs which is something I would really appreciate the authors for. This personally helped me go deeper in the paper and understand the work more throoughly.

**Weaknesses:**

I have the following concerns regarding the paper:
1. For LLM formalizer experiments, the solver being used for a given task also matters. Logic LM [1] and, [2] both use constraints library which is more natural for solving problems with planning/CSP related tasks i.e, Meeting Planning, Trip Planning and Calendar Planning. I think this experiments would be helpful.
2. Both [1] and [2] which use LLM as a formalizer have been proven to be effective in logical reasoning tasks and planning tasks on Multiple datasets such as ProntoQA [3], FOLIO [4], Proofwriter [5] and these are in my opinion more suitable datasets for evaluating LLMs as a formalizer paradigm, Can you please explain the choice of the datasets that you have choosen and why do LLMs as a formalizer peform much better [3], [4], [5] than on the datasets that you have chosen? Is it because of the solver that has been chosen that i have mentioned earlier in 1.?
3. Z3 solver is more suitable for logical reasoning tasks and using LLM as a formalizer with Z3 would be also be more appropriate on Logical Reasoning datasets. Would you be able to confirm if the results are consistent on a logical reasoning dataset for example [6]







[1] LOGIC-LM: Empowering Large Language Models with Symbolic Solvers for Faithful Logical Reasoning

[2] FCoReBench: Can Large Language Models Solve Challenging First-Order Combinatorial Reasoning Problems?

[3]  Language models are greedy reasoners: A systematic formal analysis of chain-of-thought

[4] FOLIO: natural language reasoning with first-order logic.

[5] Proofwriter: Generating implications, proofs, and abductive statements over natural language.

[6] LogicBench: Towards Systematic Evaluation of Logical Reasoning Ability of Large Language Models

**Questions:**

Please address the weakness

---

> ### Author Response · Authors · 2025-11-19
>
> We would like to thank the reviewer for their thoughtful and detailed feedback. We appreciate the opportunity to clarify our experimental design decisions and the relationship between dataset choice, solver choice, and the performance of LLM-as-formalizer approaches.

---

> ### Author Response · Authors · 2025-11-19
> **Choice of solver**
>
> We agree that the choice of solver can significantly influence downstream task performance. As the reviewer rightfully points out, prior work such as [1] and [2] uses python-constraint for Logical Deduction tasks. Yet, both papers rely on Z3 for analytical reasoning and several other benchmarks. We similarly chose Z3 as it is a better fit for our evaluation needs, primarily for 2 reasons:
> - Expressivity: Z3 is a general-purpose, SMT-based solver that handles a wider range of logical and arithmetic constraints than python-constraint.
> - Optimization: Several of our tasks require optimizing objectives (e.g., minimizing travel time or maximizing preference), which Z3 supports natively and python-constraint does not.
> In addition, Z3 is efficient and, when correctly written, reliably solves all tasks in our benchmark, helping ensure that performance differences reflect the quality of the LLM’s formalization rather than solver limitations. Z3 also provides solver statistics that we use in our analysis. For these reasons, we opted to use Z3 throughout our experiments.
>
> That said, we will follow the reviewer's suggestion to use the python-constraints library as the third formalism/tool in our study. We will report back the results once we have them.

---

> > ### Comment · Reviewer_p6Ft · 2025-11-21
> >
> > I would like to acknowledge the authors comments and I want to let you know that I would comment fully once you come back with the results of the experiments using python constraints library

---

> ### Author Response · Authors · 2025-11-19
> **Choice of datasets and domains**
>
> The short answer is that we identify **first** the problem to solve (real-life CSP) and **then** the appropriate tool (Python or Z3). Overall, we agree with the assessment that LLM-as-formalizer is more effective in prior datasets ([1], which includes most of [2-6]) than in ours (i.e, Meeting Planning, Trip Planning and Calendar Planning). We have made the same claim in line 34-36 while citing works in the planning domain. However, the key research question of our work is not “**on which tasks** is LLM-as-[Z3/any solver]-formalizer the most effective”, but rather “**whether** LLM-as-formalizer is effective on real-life CSP tasks” (line 46). What we did not emphasize enough is the motivation for those tasks themselves [7,8,9]. The datasets we consider are thus much linguistically richer, requiring common-sense inference along with algorithmic calculation. While we do not directly compare with logical reasoning datasets of a more synthetic nature (like ProofWriter), Section 5.3 shed light on translation errors that may be more rampant when the problem description is more natural.

---

> ### Author Response · Authors · 2025-11-19
> **References**
>
> Reviewer's references above, plus:
>
> [7] Planning Anything with Rigor: General-Purpose Zero-Shot Planning with LLM-based Formalized Programming (ICLR 2025)
>
> [8] TravelPlanner: A Benchmark for Real-World Planning with Language Agents (ICML 2024)
>
> [9] NATURAL PLAN: Benchmarking LLMs on Natural Language Planning

---

> ### Author Response · Authors · 2025-11-24
> **Results of python-constraint**
>
> We again thank the reviewer for the proposal to consider another tool/formalism which is indeed appropriate for the CSP tasks. We have finished extra experiments with two additional models DeepSeek-V3 and GPT-5 on all 4 domains, with the results shown below:
> | Calendar planning | Python | PyCon | SMT |
> |-------------------|--------|-------|-----|
> | DeepSeek-V3       | 91     | 95    | 96  |
> | gpt-5-2025-08-07  | 97     | 97    | 100 |
>
> | Meeting planning  | Python | PyCon | SMT |
> |-------------------|--------|-------|-----|
> | DeepSeek-V3       | 54     | 21    | 49  |
> | gpt-5-2025-08-07  | 17     | 8     | 7   |
>
> | Trip planning     | Python | PyCon | SMT |
> |-------------------|--------|-------|-----|
> | DeepSeek-V3       | 6      | 0     | 2   |
> | gpt-5-2025-08-07  | 85     | 75    | 42  |
>
> | ZebraLogic        | Python | PyCon | SMT |
> |-------------------|--------|-------|-----|
> | DeepSeek-V3       | 82     | 63    | 85  |
> | gpt-5-2025-08-07  | 95     | 100   | 90  |
>
> In these 8 combinations, the formalizing performance using python-constraint (PyCon) has an average rank of around 2 within the 3 formalisms, often neither the worst nor the best. We note that PyCon is more specific than Python but less so than SMT. Its lack of search features means it need to rely on the generated Python code to perform the optimization. Similar to the pure Python approach, this leads to up to 30 minutes of run time on some examples in our datasets, while we set a cut-off of 2 minutes. We will discuss this further in the paper, too.
>
> Examining 10 failure cases for DeepSeek-V3 on Meeting planning with PyCon, we observe the following:
> - Wrong duration/availability - 50% (5/10): Itineraries stretch meetings far past allowed windows or change required start times (examples: 115, 131, 180, 286, 577).
> - Missing/extra or wrong attendee coverage - 20% (2/10): The model drops or adds extra required meetings or swaps the wrong person/location (125, 952).
> - Travel-buffer mistakes - 10% (1/10): Start times ignore the mandated travel time difference, so the meeting begins before arrival time (118).
> - Runtime/codegen errors - 20% (2/10): Scripts either crash or hang until the 120-second timeout (203, 403).
>
> These observations are generally in line with our existing conclusions in the paper. To sum up, we will:
> 1. Add this set of experiments to the manuscript, along with analysis and discussion on run-time
> 2. Discuss the expressivity and features of each of three formalisms
> 3. Add the motivation of choosing CSP tasks as the evaluation target to Section 1&2, citing related work above
>
> We look forward to hearing of the reviewer's concern has been addressed.

---

### Author Response · Authors · 2025-11-30
**Summary of all concerns addressed**

We thank the reviewers and AC for their efforts. While the reviewers were not given a chance to respond to us, we are confident to have fully addressed all concerns. To help the AC navigate the discussion, we provide a summary below.
- Reviewer p6Ft's **(rating of 4)** primary suggestion is to also consider the `python-constraints` library in addition to the existing tools we have considered (Python and Z3) for CSP. We followed this suggestion, posted the full results, analyzed them, and noted that they do not change our conclusions in any way.
- Reviewer jBmf's **(rating of 4)** primary concern is more analysis into models' answer rate and scale, in addition to the existing research questions we have answered (section 5.1, 5.2, 5.3). We provided additional experimental results regarding the former, and argued the inability to scale larger due to the limitation of the standard benchmarks.
- Reviewer R8gZ's **(soundness, presentation, and contribution of 3)** primary concern is the our motivation rather than questioning our work itself. We clearly described a quick clarification to be added to the camera-ready that explains why we choose to focus on the CSP domains.
- Reviewer mhMi **(rating of 8)** is strongly favorable of our work. We took their suggestion to perform one further analysis, which again does not change our key conclusions.

---

### Meta-Review · Area_Chair_m7Nk · 2026-01-05

**Summary:**

This work investigates the performance of LLM-as-solver and LLM-as-formalizer. As opposed to the belief that LLM-as-solver is weaker than LLM-as-formalizer, the authors showed that LLM-as-formalizer may be subject to critical flaws such as mistranslation of constraints and insufficient reasoning.

During the rebuttal process, the authors presented results on different solvers, clarified the scope of the work and offered explanations for a range of questions the reviewers had.

Main outstanding concerns include the scope of the datasets, experiments with informative solvers and the soundness of findings. Overall, I think the scope “Are LLMs better formalizers than solvers on complex problems” does not match the level of comprehensiveness in the experiments and the associated findings. Hence I recommend rejection.

**Reviewer Concerns:**

Reviewer p6Ft raised concerns on the lack of experiments for solvers (addressed concern) and the choice of datasets (outstanding concern). The authors conducted more experiments with PyCon and provided analyses to shed more light on the failure modes. I believe this has addressed the concern on experiments. For the concern on datasets, authors explained that their focus is whether LLM-as-formalizer is effective rather than on which tasks it is effective. I think it’d be difficult to separate these two questions since evaluating the ability requires a series of tasks. It would be helpful to understand the nature of the tasks where LLM-as-formalizer works or does not work well. I’m also not fully convinced that the CSPs considered in the paper are real-life tasks. Generic users probably would not use Zebra Logic.


Reviewer jBmf raised concerns on soundness and contributions, including the interpretation and presentation of the findings, as well as the lack of considerations on problem complexity and scalability. The authors argued that it is sufficient to treat LLM-as-solver as a baseline and it is not in scope to augment existing datasets. Since the paper is about “Are LLMs better formalizers than solvers on complex problems”, I believe it is reasonable to analyze LLM-as-solver in the same way as LLM-as-formalizer, and expanding on “complex problems” also seems necessary. Alternatively I suggest the authors scope down this paper in writing to ensure their scope of experiments match the paper writing, and the claims should also be clearer in terms of what “complex problems” mean and for which level of complexity LLMs are better formalizers or solvers.

Reviewer R8gZ raised concerns on whether CPSs are the right problems to compare LLM-as-solver v.s. as-formalizer and lack of exploration of more advanced LLM-as-formalizer. The authors didn’t seem to directly address the first concern and copied a reply to Reviewer p6Ft. For the second concern the authors argued that there is no commonly agreed-upon sophisticated LLM-as-formalizer.

Reviewer mhMi raised concerns on the lack of deeper insights on cases with more informative solvers and clarity of figures. For the 1st concern, the authors “are unable to include those expansions due to scope limit”, but the authors agreed to adjust the figures.

**Reviewer Scores:**

I believe the reviewers would not have changed their scores given that each had outstanding concerns.

---

### Decision · Program_Chairs · 2026-01-26

Reject